# Framework to construct and interpret latent class trajectory modelling

Hannah Lennon,[1,2] Scott Kelly,[3] Matthew Sperrin,[2] Iain Buchan,[2] Amanda J Cross,[4] Michael Leitzmann,[5] Michael B Cook,[3] Andrew G Renehan[1,2,6]

[1]Division of Cancer Sciences, School of Medical Sciences, Faculty of Biology, Medicine and Health, University of Manchester, Manchester, UK
[2]MRC Health eResearch Centre (HeRC), Division of Informatics, Imaging and Data Sciences, University of Manchester, Manchester, UK
[3]Division of Cancer Epidemiology and Genetics, National Cancer Institute, National Institutes of Health, Bethesda, Maryland, USA
[4]Department of Epidemiology and Biostatistics, Imperial College, London, UK
[5]Department of Epidemiology and Preventive Medicine, University of Regensburg, Regensburg, Germany
[6]Manchester Cancer Research Centre, NIHR Manchester Biochemical Research Centre, University of Manchester, Manchester, UK

**Correspondence to**
Dr Hannah Lennon;
lennonh@fellows.iarc.fr and
Prof. Andrew G Renehan;
andrew.renehan@manchester.ac.uk

## ABSTRACT

**Objectives** Latent class trajectory modelling (LCTM) is a relatively new methodology in epidemiology to describe life-course exposures, which simplifies heterogeneous populations into homogeneous patterns or classes. However, for a given dataset, it is possible to derive scores of different models based on number of classes, model structure and trajectory property. Here, we rationalise a systematic framework to derive a 'core' favoured model.

**Methods** We developed an eight-step framework: step 1: a scoping model; step 2: refining the number of classes; step 3: refining model structure (from fixed-effects through to a flexible random-effect specification); step 4: model adequacy assessment; step 5: graphical presentations; step 6: use of additional discrimination tools ('degree of separation'; Elsensohn's envelope of residual plots); step 7: clinical characterisation and plausibility; and step 8: sensitivity analysis. We illustrated these steps using data from the NIH-AARP cohort of repeated determinations of body mass index (BMI) at baseline (mean age: 62.5 years), and BMI derived by weight recall at ages 18, 35 and 50 years.

**Results** From 288 993 participants, we derived a five-class model for each gender (men: 177 455; women: 111 538). From seven model structures, the favoured model was a proportional random quadratic structure (model F). Favourable properties were also noted for the unrestricted random quadratic structure (model G). However, class proportions varied considerably by model structure—concordance between models F and G were moderate (Cohen κ: men, 0.57; women, 0.65) but poor with other models. Model adequacy assessments, evaluations using discrimination tools, clinical plausibility and sensitivity analyses supported our model selection.

**Conclusion** We propose a framework to construct and select a 'core' LCTM, which will facilitate generalisability of results in future studies.

## INTRODUCTION

In many epidemiological studies, a risk factor is measured at a single point in time and related to the subsequent development of disease under the assumption that a single 'one-off' measure is an approximation for that exposure over a long time. Thus, baseline measurement of body mass index (BMI) is associated with subsequent development of common disease like cardiovascular disease,[1] diabetes,[2] several cancers[3] and all-cause

### Strengths and limitations of this study

► We developed a systematic approach, with rationale for each of eight steps, to derive a latent class trajectory model of favoured number of classes and 'core' model structure specification.

► The results presented here are based on modelling data from a large well-characterised US cohort, allowing the derivation of numerically meaningful subpopulations (ie, classes) with distinct phenotypes.

► Compared with 'one-off' body mass index categorisation, latent class trajectory modelling offers additional phenotypic information and opportunities to identify and intervene early in subpopulations with adverse trajectories.

► While we described multiple diagnostic tests, ultimately model selection was based on case study appropriate model interpretation (eg, model adequacy, discrimination, clinical plausibility and sensitivity analyses) by a multidisciplinary research team.

mortality.[4] This approach is crude, and many investigators seek to use alternative methods that might better capture long-term risk factor exposure termed *life-course analysis*. There are widely used examples that capture cumulative exposure, such as pack-years for smoking and lung cancer, but the assumption that incidence rate is proportional to total lifetime dose is questionable.[5] Many other life-course models simply extract features for use in standard regression approaches, for example, a weight change over time. A more sophisticated approach, which takes account of within-individual correlations, is *mixed-effect modelling*, but this is difficult to interpret for public health implementation. An extension of this approach is the use of *latent classes*, also termed growth mixture models.

Latent class trajectory modelling (LCTM) simplifies heterogeneous populations into more homogeneous clusters or classes. From these, one can potentially include random effects to allow for individual variation within these classes. These models have a long history in the criminology[6] and psychology[7] literatures, and now, are increasingly reported

in the human epidemiology literature (eg, disentangling the heterogeneity of childhood asthma[8]). Of relevance to this paper, LCTM has been used in association studies of repeated BMI measures with the following endpoints: all-cause mortality,[9] cancer incidence (multiple cancer types[10], gastro-oesophageal,[11] prostate[12]) and cancer mortality.[12] The LCTM has three general advantages compared with using 'one-off' exposure determinations: first, it better informs aetiological associations by deeply phenotyping certain 'at risk' subpopulations; and second, LCTM offers a public health strategy to identify early divergent adverse trajectories as potential intervention targets. Some researchers additionally argue that LCTM is well equipped for future forecasting and new patient generalisations in prediction models, as it handles data following a different predictable pattern from that learnt by the model.[13] Third, the trajectory approach allows a better understanding of the causes of between-individual variation in certain features (eg, weight variation over age), by analysing the trajectory as an outcome rather than exposure.

However, LCTM is a complex form of modelling and requires several different structure assumptions.[14] Although firmly acknowledged in the GRoLTS-Checklist: Guidelines for Reporting on Latent Trajectory Studies,[15] structure-related assumptions have not been systematically evaluated. For many exposures of interest, typically two to seven classes might be described and, as detailed later, at least seven model structures might be fitted, with and without linear curve properties, such that it is possible to derive greater than 80 different models. Thus, reported differences between studies using latent class modelling might reflect different modelling assumptions rather than true differences between populations. To facilitate the generalisability of results in future studies, here, we propose a framework to construct and select a 'core' LCTM, using an example of repeatedly determined BMI across adulthood in the National Institutes of Health (NIH)-AARP Diet and Health Study cohort. For exposure-disease outcome association analyses, current approaches generally use two stages: first, LCTM, followed by standard association modelling. The framework described here is limited to the first stage.

## METHODS
### Cohort
The NIH-AARP Diet and Health Study is a US cohort recruited from 1995.[16] A baseline medical and lifestyle questionnaire, including self-reported weight and height, was returned by 566 398 participants (aged 50–71 years; mean age: 62.5 years). An additional risk factor questionnaire was mailed in 1996 and completed by a subcohort of 327 860, of whom 288 993 (177 455 men and 111 538 women) provided recall weight for all four time points: ages 18, 35 and 50 years. Derived BMIs at baseline and these ages (assuming constant height) form the data in the present analysis. We excluded participants with

extreme BMI values ($<15$ or $>70 \,kg/m^2$) recorded at any time point. Means and SD for derived recalled BMI distributions are representative of BMI distributions for historical period-equivalent US populations.[17]

### Latent class trajectory modelling
We developed an eight step framework (table 1) modelling BMI as a function of age. Latent classes were used to identify subgroups of participants with distinct trajectories (detailed mathematical equations in online supplementary material p2).[18] We used maximum likelihood approaches to fit the model with the '*hlme*' function from 'lcmm' library[19] in the R software environment (V.3.2.1) and cross-checked results using the '*PROC TRAJ*' function in 'SAS traj' library (SAS Institute, Cary, North Carolina, USA)[20] (online supplementary table S1).

### Step 1
We initially constructed a scoping model provisionally selecting the plausible number of classes based on available literature; in the context of BMI trajectories, we used $K=5$ classes as reported elsewhere.[10 12] We built models for both genders, as BMI patterns of lifetime changes differ for men and women.[21] To determine the initial working model structure of random effects, we followed the rationale of Verbeke and Molenbergh[22] and examined the shape of standardised residual plots for each of the five classes in a model with no random effects. If the residual profile could be approximated by a flat, straight line or a curve, then a random intercept, slope or quadratic term, respectively, were considered. Preliminary plots suggested preference for a quadratic random effects model (supplementary figure S1).

### Step 2
We refined the preliminary working model from step 1 to determine the optimal number of classes, testing $K=1$–$7$. The number of classes chosen was based on the lowest Bayesian information criteria (BIC).

### Step 3
We further refined the model using the favoured $K$ derived in step 2, testing for the optimal model structure. We tested seven models (detailed in online supplementary table S2), ranging from a simple fixed effects model (model A) through a rudimentary method that allows the residual variances to vary between classes (model B) to a suite of five random effects models with different variance structures (models C–G).

### Step 4
We then performed a number of model adequacy assessments. First, for each participant, we calculated the posterior probability of being assigned to each trajectory class and assigned the individual to the class with the highest probability. An average of these maximum posterior probability of assignments (APPA) above 70%, in all classes, is regarded as acceptable.[6] We further assessed model adequacy using odds of correct classification, mismatch

**Table 1** Framework of eight steps to construct a latent class trajectory model

| Step | Step description | Criteria for selection |
|---|---|---|
| 1 | Scope model by provisionally selecting a plausible number of classes based on available literature and the structure based on plausible clinical patterns. | Examine linearity of the shape of standardised residual plots for each of the classes in a model with no random effects. |
| 2 | Refine the model from step 1 to confirm the optimal number of classes, typically testing K=1–7 classes. | Lowest Bayesian information criteria value. |
| 3 | Refine optimal model structure from fixed through to unrestricted random effects of the model using the favoured K derived in step 2. | |
| 4 | Run model adequacy assessments as described in online supplementary table S3 including posterior probability of assignments (APPA), odds of correct classification (OCC) and relative entropy. | ► APPA: average of maximum probabilities should be greater than 70% for all classes. <br> ► OCC values greater than 5.0. <br> ► Relative entropy values greater than 0.5. |
| 5 | Investigate graphical presentation | ► Plot mean trajectories across time for each class in a single graph. <br> ► Plot mean trajectories with 95% predictive intervals for each class (one class per graph). <br> ► Plot individual class 'spaghetti plots' across time for a random sample. |
| 6 | Run additional tools to assess discrimination including Degrees of separation (DoS) and Elsensohn's envelope of residuals | ► DoS greater than zero. <br> ► Envelope of residuals is assessed in plots by observing clear separations between classes. |
| 7 | Assess for clinical characterisation and plausibility. | ► Tabulation of characteristics by latent classes. Are the trajectory patterns clinically meaningful? Perhaps, consider classes with a minimum percentage of the population. <br> ► Are the trajectory patterns clinically plausible? <br> ► Concordance of class characteristics with those for other well-established variables. |

Continued

**Table 1** Continued

| Step | Step description | Criteria for selection |
|---|---|---|
| 8 | Conduct sensitivity analyses, for example, testing models without complete data at all time points. | General assessment of patterns of trajectories compared with main model. |

scores and entropy, $E_k$ (detailed in online supplementary table S3). These diagnostic tools assist in model selection.[6 23] In some examples, BIC values may decrease as more groups and parameters are added reflecting model overfit. Therefore, the BIC value might not always provide the optimum selection criteria, and model selection must balance between meaningful trajectories, model parsimony and model adequacy. For example, if the model adequacy measures are strongly violated, one might go back to steps 2 and 3 and consider a different model with a higher BIC value. We selected an optimal model structure using the lowest BIC value and satisfactory values from the model adequacy assessments and referred to the outcome of steps 1–4 as the *favoured model*. To assess the interpretability of the resulting classes, we investigated characteristics of lifestyle behaviours of the favoured model such as smoking, alcohol consumption and physical activity.

### Step 5

We used three graphical presentation approaches. The conventional approach is to plot mean trajectories with time encompassing each class. Alternatives include the use of mean trajectory plots with 95% predictive intervals for each class, which displays the predicted random variation within each class, or to plot individual level 'spaghetti plots' with time (eg, a random sample of participants), which allows the reader to observe the patterns of changes within classes.

### Step 6

We assessed model discrimination, including degrees of separation, $DoS_K$,[24 25] and Elsensohn's envelope of residuals.[25] To describe the separation of latent trajectory curves, a multivariate Mahalanobis distance was used. Peugh and Fan[26] argue that it is reasonable to speculate that identification of heterogeneous latent trajectories is facilitated by large statistical separation distance among the subpopulations. Thus, larger values of $DoS_K$ indicate the mean trajectories are well separated, while $DoS_K$ equal to zero is the special case when all mean trajectories are identical. If the $DoS_K$ value is small, then one might consider a model with fewer classes.

To check structure assumptions in fixed effects latent class models, Elsenhohn *et al*[25] plotted the local SD of the residuals against time. We extended this method to random effects models: first, computing the observed residuals for each participant; and second, computing the class-specific and time-specific weighted local variance

**Table 2** Number of classes (K=1–7) using random effects quadratic structure model F (proportional covariance structure) by gender

| Model | K | Number of parameters | BIC | Proportions per class % | | | | | | |
| | | | | Class I | Class II | Class III | Class IV | Class V | Class VI | Class VII |
|---|---|---|---|---|---|---|---|---|---|---|
| **Men** | | | | | | | | | | |
| Model F | 1 | 10 | * | 100 | | | | | | |
| | 2 | 15 | 3 324 009 | 83 | 17 | | | | | |
| | 3 | 20 | 3 310 908 | 62 | 32 | 3 | | | | |
| | 4 | 25 | 3 324 128 | 100 | 0 | 0 | 0 | | | |
| | 5 | 30 | **3 301 301** | 68 | 25 | 4 | 3 | 0.4 | | |
| | 6 | 35 | * | | | | | | | |
| | 7 | 40 | * | | | | | | | |
| **Women** | | | | | | | | | | |
| Model F | 1 | 10 | * | 100 | | | | | | |
| | 2 | 15 | 2 195 386 | 86 | 14 | | | | | |
| | 3 | 20 | 2 179 080 | 58 | 34 | 8 | | | | |
| | 4 | 25 | 2 179 137 | 100 | 0 | 0 | 0 | | | |
| | 5 | 30 | 2 169 791 | 41 | 32 | 21 | 4 | 2 | | |
| | 6 | 35 | * | | | | | | | |
| | 7 | 40 | * | | | | | | | |

Results from random effects quadratic structure model G (unrestricted) by gender are shown in supplemental material S2.
*Models failed to converge.
BIC, Bayesian information criteria.

of the residuals, with weights being the posterior probabilities of individual belonging to a class. We plotted the upper and lower boundary values of the local SD of the residuals around the mean values for each class. The resulting shape indicates the appropriateness of the model assumptions, where non-parallel boundaries indicate heteroscedasticity of residuals suggesting poor model fit, and differing interval widths suggest that across class variability may not be fully accounted for.

### Step 7

We assessed for clinical characterisation and plausibility using four approaches: (1) assessing the clinical meaningfulness of the trajectory patterns, aiming to include classes with at least 1% capture of the population; (2) assessing the clinical plausibility of the trajectory classes; (3) tabulation of characteristics by latent classes versus conventional categorisations; and (4) concordance of class membership with conventional BMI category membership using the kappa statistic (as LCTM is an unsupervised learning approach, we computed $k$ for all possible combinations and selected the optimal $k$).

### Step 8

We conducted sensitivity analyses, in this example, with individuals with at least two and three BMI values, as LCTMs are flexible enough to deal with different observation times between participants.

### Patient and public involvement

No patients and or public were involved with this manuscript.

### Statistical algorithms

All R and SAS codes used to implement these tools are available via the authors and can be downloaded from www.github.com/hlennon/LCTMtools.

## RESULTS
### Number of classes

From the preliminary working model of a quadratic random effects model, model F (proportional covariance structure), we derived BICs for up to seven classes: three of the class models failed to converge in men and women. Table 2 reports that the lowest BIC was obtained with five classes in men and women, confirming our initial working model. The proportions by class in men were 68.1%, 25.0%, 3.8%, 2.7% and 0.4%, and in women, proportions by class were 32.6%, 41.1%, 21.1%, 3.5% and 1.7%. For model G (our second favoured model), the lowest BICs were noted for five classes in men and women (online supplementary table S2).

### Assessment for model structures

With the number of classes now selected as five, we tested the seven model structures: A–G. Table 3 reports that

**Table 3** Model adequacy assessments of latent trajectory class models based on different assumptions for $K$=5 classes, by gender in the NIH-AARP cohort

| Model | Description | BIC | Proportion per class % | Average posterior probability assignment | Relative entropy ($E_K$) | Degree of separation ($DoS_K$) |
|---|---|---|---|---|---|---|
| **Men** | | | | | | |
| A | Homoscedastic | 3476322 | 51: 22: 21: 6: 1 | 83: 84: 85: 91: 95 | 0.78 | 0.10 |
| B | Heteroscedastic | 3511646 | 41: 26: 20: 7: 6 | 85: 82: 84: 89: 86 | - | 0.15 |
| C | Random intercept | 3364856 | 68: 24: 4: 3: 2 | 90: 82: 89: 85: 82 | 0.81 | 0.34 |
| D | Random slope | 3325463 | 63: 19: 13: 3: 4 | 72: 70: 74: 83: 83 | 0.59 | 0.26 |
| E | Random quadratic, Equal | * | | | | |
| F | Random quadratic, Proportional | **3301301** | 68: 25: 4: 3: 0.4 | 81: 74: 87: 83: 74 | 0.68 | 0.36 |
| G | Random quadratic, Unrestricted | 3320005 | 56: 30: 8: 5: 0.4 | 74: 70: 80: 79: 88 | 0.63 | 0.33 |
| **Women** | | | | | | |
| A | Homoscedastic | 2289509 | 45: 36: 13: 4: 1 | 90: 84: 88: 93: 96 | 0.83 | 0.14 |
| B | Heteroscedastic | 2238444 | 32: 30: 15: 15: 8 | 87: 83: 85: 89: 91 | - | 0.05 |
| C | Random intercept | 2240681 | 52: 35: 9: 2: 2 | 89: 82: 86: 89: 91 | 0.79 | 0.21 |
| D | Random slope | 2193155 | 79: 12: 5: 2: 2 | 91: 80: 79: 86: 91 | 0.82 | 0.34 |
| E | Random quadratic, Equal | 2188224 | 95: 5: 0: 0: 0 | 49: 88: 0: 0: 0 | 0.26 | 0.05 |
| F | Random quadratic, Proportional | **2169793** | 41: 33: 21: 3: 2 | 74: 79: 80: 83:84 | 0.66 | 0.09 |
| G | Random quadratic, Unrestricted | 2187707 | 67: 23: 6: 3: 1 | 84: 77: 84:82: 87 | 0.73 | 0.34 |

*Failed to converge.
BIC, Bayesian information criteria; NIH, National Institutes of Health.
Bold values are the chosen according to the lowest value.

the lowest BIC was for model F in men and women, justifying the selection of model F in the preliminary working phase. The class sizes varied between models, with class I ranging from 41% to 68% in men and from 32% to 95% in women. The APPA for each class in model F was 0.81, 0.74, 0.87, 0.83 and 0.74 in men and 0.74, 0.79, 0.80, 0.83 and 0.84 in women, indicating a good discrimination of trajectory. The classes were well differentiated with the relative entropy, $E_K$ values ranging from 0.59 to 0.81 in men and from 0.66 to 0.83 in women.

There was moderately good concordance (unweighted and weighted) between the unstructured variance models G with model F in men ($k$: 0.57) and women ($k$: 0.65) (supplementary tables S4 and S5) but poorer concordance between the favoured models and fixed-effects models in men.

### Graphical presentation

We plotted the mean trajectories for model A, B, C, D, F and G in men and women (figure 1) illustrating the increased complexity from model A to model G. As alternatives, we plotted separately mean trajectories with 95% predictive intervals for each class, in model F (online supplementary figure S2), which displays the predicted random variation

within each of the classes with time, noting that variation was greater with the more 'complex' classes (classes IV and V compared with classes I, II and III). Spaghetti plots of individual level data illustrated that the timing and size of BMI changes characterise the classes; for example, sharp increases in BMI in early adulthood in class III but later in adulthood for class IV (online supplementary figure S3).

### Additional tools of suitability of fit

The $DoS_k$ values ranged from 0.10 to 0.36 and 0 to 0.34, in men and women, respectively (table 3). The covariances were high and in the positive direction, and therefore models with non-parallel mean trajectories lead to higher separation.

We plotted the local SD of the residuals with time and found that these were broadly homogeneous, that is, there were few parallel boundaries (figure 2). The local residuals for the rapidly obese groups in both genders are the exceptions to parallel lines, which might reflect comorbidities in this group and smaller numbers.

### Clinical assessment

Having established the favoured model, model F with five classes in both genders, we assigned descriptive labels to

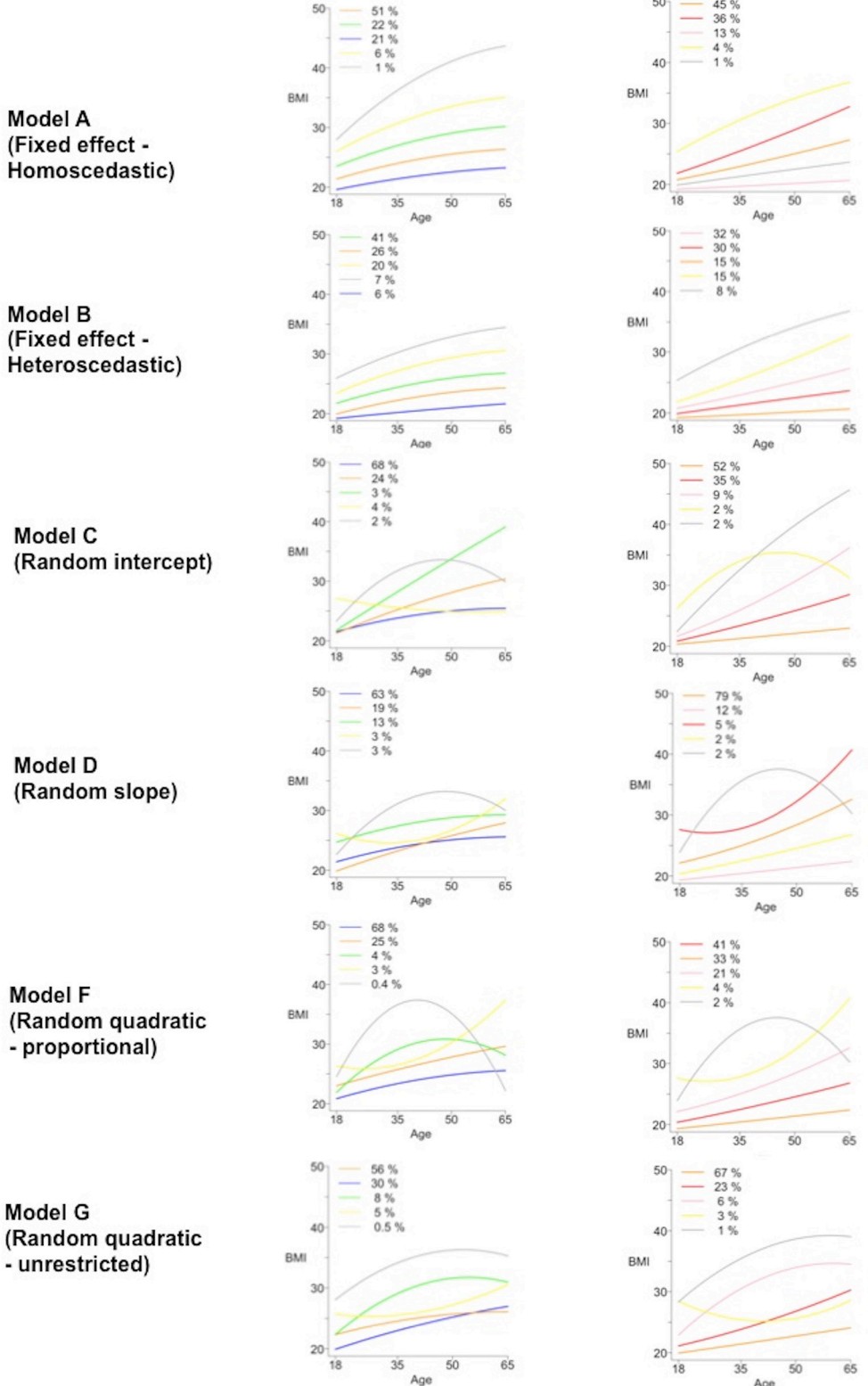

**Figure 1** BMI mean trajectories by men (left) and women (right) for models A–G. Colours are used to discriminate classes within each plot but should not be used for direct comparisons across plots. BMI, body mass index.

each respective class as follows (table 4): stable normal weight; normal weight to overweight; normal weight to obese; overweight to obese; and rapid early obesity. We noted that the proportion in the rapid early obesity (class V) was less than 1% in men. However, overall, the

proportion for class V for men and women combined was nearly 1%. Thus, we retained this class as we judged it to be clinically meaningful as follows. In both genders, there were rapid increases in obesity from early to middle adulthood, then apparent severe weight reductions. We

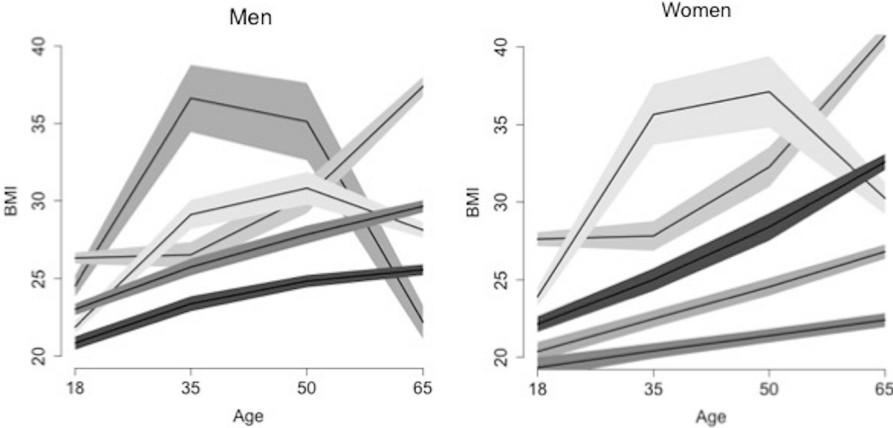

**Figure 2** Illustration of the local (Elsensohn) residual envelope plots (shown here for model B).

rationalised that this was clinically plausible, as it could be explained either by intentional (eg, bariatric surgery) or non-intentional weight loss (eg, reverse causality from development of disease).

We then tabulated the baseline characteristics according to the five classes for model F in men and women and noted patterns across the classes (table 4 and fully expanded in online supplementary tables S5 and S6). Thus, for example, for current smoking status, there were little differences in patterns by class in men: 10%, 8%, 9%, 10% and 15%, and in women: 16%, 12%, 12%, 13% and 11%. This contrasts with BMI categories: men (12%, 8%, 8%, 7% and 7%) and women (16%, 12%, 10%, 8% and 7%) (online supplementary tables S7 and S8).

Finally, we noted very poor concordance between the favoured model and conventional BMI categorisation in men (*k*: 0.18) and women (*k*: 0.52) (table 5).

### Sensitivity analyses
We tested the favoured model using a larger sample of individuals with at least three measures and found no material differences between these models, in men and women, and the main model (online supplementary figure S4).

### DISCUSSION
### Main findings
We propose an eight-step framework for the construction and selection of models derived from LCTM. We evaluated a range of model structures from fixed effect models to a set of random effects models, favouring the latter models in this case study, as they include different variance structures and more likely to reflect the natural history of changes with time in BMI distributions in different subpopulations. We showed that different model structures resulted in different classes with contrasting clinical phenotypes. We propose prespecified criteria for model selection and that the reporting of a 'core' model will facilitate generalisability of results in future studies.

### Context of other literature
To the best of our knowledge, this is the first study to systematically address structure-related assumptions in LCTMs, and their potential impact on clinically relevant endpoints—in this example, BMI trajectories. Anecdotally, there is a justifiable criticism regarding the use of LCTM models and an uncertainty of how class memberships are derived—a 'black box' effect. The proposed framework, here, encourages the opposite—a transparent stepwise approach to class and model structure selection. To enhance this process, for example, we have 'borrowed' tools developed to address to quantify uncertainty, such as entropy measures, $E$ and $E_k$, and applied them to assist assessment of model adequacy. A further modification of discrimination measurement with variance estimation has been described by Shah and colleagues[27] and might have importance for class assignment where 'yes/no' treatment decisions are required.

Variations of model A (fixed effects) have been reported in the clinical literature,[9–12] which assume no within-class variability when deriving latent classes. Interpretation in this setting is that variation from the mean trajectory is random; that is, the correlation between measurements for the same individual is explained by latent class membership. In the context of any repeated measures in the general population, this assumption might not be valid.[14] Saunders[28] argued in support of full random effects models (ie, models F and G), calling on Moffitt's theory from criminology, which recognises that '*there are distinct developmental clusters of trajectories of anti-social behaviour that are the result of divergent aetiologies*'; in other words, it is unlikely that latent classes start from a similar baseline.

The publication of the 16-item GRoLTS Checklist in 2017[15] heralded an important advance for the application of LCTM. Here, we add a framework for construction and interpretation.

### Strengths and weaknesses
The study has strengths. First, the considered and strategic workflow to optimise identification and application of

**Table 4** Latent class characteristics of 177 453* men and 111 503* women in the NIH-AARP cohort

| | Model F (favoured model) | | | | |
|---|---|---|---|---|---|
| | Class I | Class II | Class III | Class IV | Class V |
| | Stable normal weight | Normal weight to overweight | Normal weight to obese | Overweight to obese | Rapid increase to obese |
| **Men** | | | | | |
| Number, n (%) | 120 867 (68.1) | 44 383 (25.0) | 6 723 (3.8) | 4 764 (2.7) | 718 (0.4) |
| Mean (SD) BMI at 18 years | 20.75 (0.01) | 23.92 (0.02) | 21.69 (0.04) | 27.16 (0.09) | 24.76 (0.2) |
| Mean (SD) BMI at 35 years | 23.35 (0.01) | 26.72 (0.01) | 30.24 (0.04) | 26.61 (0.09) | 37.69 (0.26) |
| Mean (SD) BMI at 50 years | 24.66 (0.01) | 29.12 (0.02) | 31.34 (0.05) | 30.95 (0.12) | 35.01 (0.35) |
| Mean (SD) entry age, years | 62.88 (0.01) | 61.59 (0.03) | 62.03 (0.06) | 59.71 (0.07) | 57.73 (0.18) |
| Mean (SD) current (at baseline) BMI, kg/m$^2$ | 25.39 (0.01) | 30.84 (0.02) | 29.07 (0.05) | 35.14 (0.12) | 30.51 (0.28) |
| Mean (SD) waist circumference, cm | 94.38 (0.02) | 106.23 (0.05) | 102.73 (0.13) | 115.24 (0.28) | 105.91 (0.64) |
| Smoking, n (%) | | | | | |
| Current | 11 823 (10) | 3 761 (8) | 638 (9) | 460 (10) | 106 (15) |
| Former | 67 948 (56) | 26 898 (61) | 3 809 (57) | 3 002 (63) | 370 (52) |
| Never | 37 149 (31) | 12 130 (27) | 2 012 (30) | 1 110 (23) | 218 (30) |
| Missing | 3 946 (3) | 1 594 (4) | 264 (4) | 191 (4) | 24 (3) |
| Mean (SD) alcohol g/day | 18.69 (0.13) | 17.45 (0.21) | 15.48 (0.56) | 15.57 (0.69) | 10.22 (1.14) |
| **Women** | | | | | |
| Number, n (%) | 36 311 (32.6) | 45 832 (41.1) | 23 544 (21.1) | 3 898 (3.5) | 1 918 (1.7) |
| Mean (SD) entry age, years | 62.48 (0.03) | 62.23 (0.02) | 61.07 (0.03) | 59.61 (0.09) | 59.69 (0.12) |
| Mean (SD) BMI at 18 years | 19.3 (0.01) | 20.44 (0.01) | 22.61 (0.02) | 29.01 (0.09) | 23.97 (0.1) |
| Mean (SD) BMI at 35 years | 20.26 (0.01) | 22.53 (0.01) | 25.51 (0.02) | 28.38 (0.1) | 36.92 (0.16) |
| Mean (SD) BMI at 50 years | 21.16 (0.01) | 24.77 (0.01) | 29.52 (0.03) | 33.51 (0.13) | 37.69 (0.2) |
| Mean (SD) current (at baseline) BMI, kg/m$^2$ | 21.97 (0.01) | 26.66 (0.01) | 32.4 (0.03) | 37.47 (0.15) | 34.4 (0.19) |
| Mean (SD) waist circumference, cm | 76.19 (0.04) | 86.66 (0.05) | 97.82 (0.09) | 106.29 (0.32) | 101.96 (0.41) |
| Smoking, n (%) | | | | | |
| Current | 5803 (16) | 5660 (12) | 2733 (12) | 495 (13) | 214 (11) |
| Former | 14 173 (39) | 18 557 (40) | 9748 (41) | 1764 (45) | 731 (38) |
| Never | 15 313 (42) | 20 290 (44) | 10 417 (44) | 1507 (39) | 909 (47) |
| Missing | 1022 (3) | 1325 (3) | 646 (3) | 132 (3) | 64 (3) |
| Mean (SD) alcohol g/day | 8.07 (0.1) | 6.26 (0.08) | 4.47 (0.11) | 3.92 (0.29) | 3.4 (0.48) |
| Hormone therapy use, n (%) | | | | | |
| Ever | 13 989 (39) | 20 206 (44) | 12 215 (52) | 2241 (57) | 1092 (57) |
| Never | 22 322 (61) | 25 626 (56) | 11 329 (48) | 1657 (43) | 826 (43) |

*Exclusions include 35 women and 2 men with biologically implausible cancers from classes in proportions (2, 16, 12, 0 and 5) and (1, 0, 0, 1 and 0), respectively.
BMI, body mass index; NIH, National Institutes of Health.

**Table 5** Concurrence between BMI categories and classes in model F from the NIH-AARP cohort

| Latent class | N | BMI categories (kg/m$^2$) | | | | |
|---|---|---|---|---|---|---|
| | | <18.5 | 18.5 to 24.9 | 25.0 to 29.9 | 30.0 to 34.9 | >35.0 |
| Men | | | | | | |
| | 177453 | 875 | 53466 | 86967 | 28234 | 7911 |
| I | 120866 | 312 | 50171 | 68646 | 1736 | 1 |
| II | 44383 | 319 | 1860 | 14097 | 23472 | 4635 |
| III | 6723 | 47 | 859 | 3335 | 1943 | 539 |
| IV | 4763 | 172 | 437 | 679 | 917 | 2558 |
| V | 718 | 25 | 139 | 210 | 166 | 178 |
| | | Cohen $K_\omega$ = 0.182 (0.181 to 0.183) | | | | |
| Women | | | | | | |
| | 111503 | 1327 | 48273 | 36025 | 16245 | 9633 |
| I | 36311 | 1215 | 33371 | 1725 | 0 | 0 |
| II | 45832 | 3 | 12584 | 28636 | 4607 | 2 |
| III | 23544 | 83 | 1683 | 4777 | 10587 | 6414 |
| IV | 3898 | 14 | 382 | 519 | 614 | 2369 |
| V | 1918 | 12 | 253 | 368 | 437 | 848 |
| | | Cohen $K_\omega$= 0.52 (0.51 to 0.53) | | | | |

$K_\omega$ weighted.
BMI, body mass index; NIH, National Institutes of Health.

latent classes provides for a more robust and transparent application of these models in epidemiology. Second, the results presented are based on modelling data from a large well-characterised US cohort, therefore allowing the derivation of numerically meaningful subpopulations (ie, classes) with distinct phenotypes. We uniquely used averaged kappa values to demonstrate that the LCTM-derived subpopulations are markedly different to those derived from a 'one-off' BMI determinations. In turn, BMI trajectories are more likely to reflect normal clinical practice of considering a 'weight history'. Third, we extensively explored different model selections and adequacy tools, and described extensions to other tools, to supplement model interpretation. Fifth, tofurther supplement model interpretation, we embedded this project within a multidisciplinary research team including data scientists, statisticians, clinicians and epidemiologists—an approach echoed elsewhere.[29] Finally, we have made the statistical algorithms freely available.

There are several study weaknesses. First, LCTMs currently only considers trajectories of one risk factor at a time. Second, there were only four time points in the AARP such that it was not possible to assess weight cycling. Third, while we described multiple diagnostic tests, ultimately model selection was based on case study appropriate model interpretation (eg, model adequacy, discrimination, clinical plausibility and sensitivity analyses) as well as likelihood-based model fit criteria.[18] Some discussion on statistical power and efficiency is warranted. The objective of model selection is a trade-off between efficiency and validation with the aim of summarising

distinct features of the data as parsimonious as possible and not just the maximisation of model fits.[6] For example, in a hypothetical scenario, putting too much emphasis on the validity of a model in which 10 classes provide the best model fit is questionable if 3 of the 10 classes each include less than 0.5% of the population and do not show markedly different characteristics.

### Clinical implications and future research
We showed that different model structures resulted in different classes with contrasting clinical phenotypes. Thus, for example, it is well recognised that the proportions of current smokers decreases with increasing BMI categories. However, this 'trend' is not observed across the latent classes derived in our favoured model F, suggesting that the clinical characteristics derived from the LCTM differed from those derived from conventional categorisation approaches. Thus, compared with 'one-off' BMI categorisation, LCTM offers additional phenotypic information.

For future research, improving the construction, interpretation and reporting of LCTM (advocated here) is hugely important as the LCTM approach has opportunities to identify and intervene early in subpopulations with adverse trajectories. This approach is analogous to the well-held public strategy of using childhood growth charts to identify and intervening in young children failing to thrive. Thus, in the example of BMI, remembering that 80% of obese adults were not obese in childhood,[30] future LCTM studies might identify (new) individuals in their 20s or early 30s on adverse trajectories towards later

adulthood obesity. This strategy is a new methodological paradigm, as the repeated measurement of a risk factor (here, BMI) becomes a clinically relevant endpoint rather than just an exposure.

**Acknowledgements** We acknowledge the generous funding from Cancer Research UK National Awareness and Early Detection Initiative.

**Contributors** HL, MS, MBC and AGR conceptualised the paper. HL, MS and SK designed the statistical approaches; HL performed the modelling. AJC and ML facilitated data access and interpretation of the AARP data. All authors contributed to data interpretation; IB and AGR put modelling into clinical context.

**Funding** This research received no specific grant from any funding agency in the public, commercial or not-for-profit sectors.

**Competing interests** AGR has received lecture honoraria from Merck Serona and Janssen-Cilag and independent research funding from Novo Nordisk. All other authors have no conflicts of interest to declare.

**Patient consent** Not required.

**Ethics approval** NCI SSIRB (Special Studies IRB).

**Provenance and peer review** Not commissioned; externally peer reviewed.

**Data sharing statement** No additional data available.

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
