## [Reviewer comments · BMJ Open]

ARTICLE DETAILS

TITLE (PROVISIONAL)	A framework to construct and interpret latent class trajectory modelling
AUTHORS	Lennon, Hannah; Kelly, Scott; Sperrin, Matthew; Buchan, Iain; Cross, Amanda; Leitzmann, Michael; Cook, Michael; Renehan, Andrew

VERSION 1 – REVIEW

REVIEWER	Mingyang Song Harvard Medical School, USA
REVIEW RETURNED	11-Dec-2017

GENERAL COMMENTS	The study presents a systematic framework for constructing and interpreting latent class trajectory models (LCTM). This work is very timely as the LCTM is becoming increasingly popular for studying the life-course impact of exposures on chronic outcomes. The recommendations yielded from this work will be of great help to guide researchers to select appropriate models to address their substantive questions of interest. The study is very well performed and the results are carefully interpreted. I have a few major and minor comments for the authors to consider: Major questions: 1. It seems that adding random effects to the model increases the flexibility and the capability to identify the extreme trajectory groups (eg, the weight loss group in the example study). It would be interesting to see how well this may have captured the real change in the study sample (ie, how many subjects in the group 5 really lost weight?) Including the BMI data in each age across each of the trajectory groups to table 4 may be helpful. Also, is it possible that identifying these trajectory groups may somewhat sacrifice the ability to dissect the dominant patterns in the population. For example, in the model F of the case study, the weight loss group only includes 0.4% of men, whereas 68% men are classified into the weight-gain group, which presumably may comprise individuals who demonstrate a large variability in their weight gain.2. Related to the prior point, while the importance of model interpretation for selecting the most appropriate analytic model is recognized in the discussion, it does not seem to be implemented in the case study, because BIC was used as the only criterion for model selection.3. Convergence issue: As the authors mentioned, sometimes the model won't converge, especially when a large number of groups with flexible-effects models are specified in contrast with the limited sample size and small number of time points. It would be helpful if the authors can comment on potential causes and solutions to address the convergence issue. Would it be better to reduce the number of groups or lower the flexibility of the model? Also, surprisingly, in the case study, model E, which is less flexible than model F, appeared not to converge in men and perform poorly in women. Thus, it almost seems that it is by chance that model F can converge and give the lowest BIC. Minor questions:
--

	 1. Introduction: It may be good to also point out the value of the trajectory approach in better understanding the causes of between-individual variation in certain features (eg, weight variation over age), by analyzing the trajectory as an outcome rather than exposure. 2. Methods: the online codes are not accessible. 3. Statistical power: should it be considered when selecting different models? Is there any effort to help researchers optimize the validity-efficiency balance when choosing the trajectory models? 4. The last point in the "strengths and limitations of this study" panel is very generic and not directly related to the primary content of the current study. It would be good to consider the limitations of the proposed framework rather than the substantive analysis. 5. Page 6: the "Step 3" paragraph, the reference to Table S2 should be S1. 6. Ref 19 is irrelevant and can be removed.
--	--

REVIEWER	Nilesh Shah University of Pittsburgh USA
REVIEW RETURNED	21-Dec-2017

GENERAL COMMENTS	I find it hard to follow the main objective of this paper. The title suggests that it concerns a framework for future studies with latent class models. The main conclusion does not answer this question. Using different models is an important part of exploring data, and it would be a bit simple to choose one model and postulate this as a framework. The trajectories are based upon self reported BMI, and also recalled BMI from years back. These are limitations which should be mentioned. Also it is mentioned that the kappa statistic may not be the best criterium to use in latent class modelling, but it is not mentioned in the methods that this value is used in any way. Furthermore I would reduce the amount of information and tables shown, for example figure FS3 seems redundant to me, it makes it difficult to find the focus of the paper. What happened to missing data? Is the paper reported according to STROBE or CONSORT guidelines? It is not explained in the methods that lifestyle behaviours were investigated, yet these are presented in the tables. Strengthen the choice made for building models for men and women separately. No confounding effects of gender? Also is there a difference in the outcomes between men and women? Overall: the manuscript needs clarification of the main objective and what was done to come to the conclusions.
---

REVIEWER	Nilesh Shah University of Pittsburgh USA
REVIEW RETURNED	21-Dec-2017

GENERAL COMMENTS	The authors outline a systematic framework and procedure for building latent class trajectory models, and demonstrate their approach by analyzing BMI data. Given the varying results that trajectory analyses can yield given different assumptions, it is important to consider decisions that lead to optimal models. The authors lay out a process to help analysts determine how to select the optimal model for their given data. This is a good and necessary step toward standardizing these types of analyses. Comments: Step 3 of the procedure is not clear. It seems as though the text does not correspond with the table referenced (Table S2). The authors may be referring to Table S1. Please include the BIC for each of these models so that it is clear how you are choosing the
---

	optimal model. In Step 4, the authors describe a series of model adequacy tests, and mention that if these are violated, one may return to a previous step and select a new model with a higher BIC. The authors may want to discuss the trade-off between BIC efficiency and model adequacy. Is it context dependent? The authors may want to explore if the BIC Factor measure provides a better idea of k rather than BIC. How would using BIC Factor lead to different results, particularly when adjusting for many covariates? It does not appear as though the authors controlled for baseline covariates in their BMI model. How would the addition of covariates (and models with more parameters) affect the model selection process? In the discussion, the authors state uncertainty in class membership is not discussed, and that is a criticism of these models. Shah, et al formulate measures for evaluating discrimination and class membership in "Measures of discrimination for latent group-based trajectory models", Journal of Applied Statistics, 42(1) 2015. One of the strengths outlined by the authors is the large dataset they used to demonstrate their process. Is there any idea how this process holds up with smaller sample sizes?
--	---

VERSION 1 – AUTHOR RESPONSE

Reviewer #1:

Reviewer Name: Mingyang Song

Institution and Country: Harvard Medical School, USA

Please state any competing interests: None declared

The study presents a systematic framework for constructing and interpreting latent class trajectory models (LCTM). This work is very timely as the LCTM is becoming increasingly popular for studying the life-course impact of exposures on chronic outcomes. The recommendations yielded from this work will be of great help to guide researchers to select appropriate models to address their substantive questions of interest. The study is very well performed and the results are carefully interpreted.

Authors' reply: Thank you for these positive comments.

I have a few major and minor comments for the authors to consider:

Major questions:

1. It sees that adding random effects to the model increases the flexibility and the capability to identify the extreme trajectory groups (eg, the weight loss group in the example study). It would be interesting to see how well this may have captured the real change in the study sample (ie, how many subjects in the group 5 really lost weight?)

Authors' reply : In our exploratory analyses, we checked the validity of the mean trajectories in describing the classes in three ways. First, we computed the prediction intervals and displayed these in Table S2. Second, we repeatedly took random samples of size n=250 from each class and plotted the raw data trajectories within each class. This was repeated ten times for each gender and for men we displayed one sample in Table S3. Third, we looked at change in BMI across the time points and calculated the proportion of individuals who lost or gained weight, where appropriate to the mean trajectory shape.

For the example the reviewer proposed, the rapid increase to obese (class 5): there are less than 0.84% who lost any weight between early adulthood 18-35 years, and approximately 70% lost weight between 50 years and study entry. Therefore a high proportion in both genders (73 and 67%) did lose weight between 50 years and baseline age, which has range of 50.3 to 71.5 years.

Group 5	Between ages	% lost weight	% lost < 2 BMI	% lost < 5 BMI
Men, n=718	18-35 years	0.84	0.28	0
	35-50 years	68	61	47
	50-baseline age	73	47	34
Women, n=1918	18-35 years	0.47	0	0
	35-50 years	43	35	27
	50-baseline age	67	46	32

Authors' action: No specific action in revision. These details are for the reviewer.

2. Including the BMI data in each age across each of the trajectory groups to table 4 may be helpful. Authors' reply and action:

In order to keep the main manuscript concise, we displayed the mean and standard deviation of BMI at ages 18, 35 and 50 across each trajectory class in the supplementary materials (Table TS5 and TS6).

We have made these more readily accessible we have added these and revised Table T4.

Mean (sd) BMI	Model F (preferred model)					
	Class I	Class II	Class III	Class IV	Class V	
Men	Stable	normal	Normal weight	Normal weight	Overweight	to Rapid increase

	weight	to overweight	to obese	obese	to obese
Age					
18 years	20.75 (0.01)	23.92 (0.02)	21.69 (0.04)	27.1 6 (0.09)	24.7 6 (0.2)
35 years	23.35 (0.01)	26.72 (0.01)	30.24 (0.04)	26.6 1 (0.09)	37.6 9 (0.26)
50 years	24.66 (0.01)	29.12 (0.02)	31.34 (0.05)	30.9 5 (0.12)	35.0 1 (0.35)
Baseline age	25.39 (0.01)	30.84 (0.02)	29.07 (0.05)	35.1 4 (0.12)	30.5 1 (0.28)
Women					
18 years	19.3 (0.01)	20.44 (0.01)	22.61 (0.02)	29.0 1 (0.09)	23.9 7 (0.1)
35 years	20.26 (0.01)	22.53 (0.01)	25.51 (0.02)	28.3 8 (0.1)	36.9 2 (0.16)
50 years	21.16 (0.01)	24.77 (0.01)	29.52 (0.03)	33.5 1 (0.13)	37.6 9 (0.2)
Baseline age	21.97 (0.01)	26.66 (0.01)	32.4 (0.03)	37.4 7 (0.15)	34.4 (0.19)

3. Also, is it possible that identifying these trajectory groups may somewhat sacrifice the ability to dissect the dominant patterns in the population. For example, in the model F of the case study, the weight loss group only includes 0.4% of men, whereas 68% men are classified into the weight-gain group, which presumably may comprise individuals who demonstrate a large variability in their weight gain.

Authors' reply and action: The reviewer correctly points out that currently these latent class trajectory models do not take account of individuals with large variability in weight gain, typically exemplified by weight cycling. This would require more time point measurements.

We now acknowledge this in our revised paragraph on study limitations (page 13) as follows:

“Second, there were only four time points in the AARP such that it was not possible to assess weight cycling.”

4. Related to the prior point, while the importance of model interpretation for selecting the most appropriate analytic model is recognized in the discussion, it does not seem to be implemented in the case study, because BIC was used as the only criterion for model selection.

Authors' reply and action: Thank you. We now appreciate that we were not clear in our initial submission. Model F was selected using the criterion of lowest BIC along with satisfactory values from the range of other metrics we outlined in the paper; a selection of which are displayed in Table 3.

However, we recommend that the final model is not solely determined using the lowest BIC criterion, but alongside the other metrics outlined in Table T1 and the meaningfulness of the groups for our research question, e.g. associations with a clinical outcome.

This is an important point, and we have corrected it in a number of places:

Summary strengths and limitations:

“Whilst we described multiple diagnostic tests, ultimately model selection was based on case-study appropriate model interpretation (for example, model adequacy; discrimination; clinical plausibility; sensitivity analyses) by a multi-disciplinary research team”.

Page 7 Methods:

“In some examples, BIC values may decrease as more groups and parameters are added reflecting model overfit. Therefore, the BIC value might not always provide the optimum selection criteria, and model selection must balance between meaningful trajectories, model parsimony and model adequacy.”

Page 13 in the Discussion:

“The trade- off between BIC efficiency and model adequacy is summarised by Nagin as “.....*the objective of the model selection is not the maximisation of some statistic of model fit (but rather)to summarise the distinctive features of the data in as parsimonious a fashion as possible*”.

5. Convergence issue: As the authors mentioned, sometimes the model won't converge, especially when a large number of groups with flexible-effects models are specified in contrast with the limited sample size and small number of time points. It would be helpful if the authors can comment on potential causes and solutions to address the convergence issue. Would it be better to reduce the number of groups or lower the flexibility of the model? Also, surprisingly, in the case study, model E, which is less flexible than model F, appeared not to converge in men and perform poorly in women. Thus, it almost seems that it is by chance that model F can converge and give the lowest BIC.

Authors' reply and action: The difference of model E and F is that the variance-covariance matrix in E is restricted to be equal across all classes, with the interpretation that the variation within each class is the same, regardless of shape or proportion in the class.

Model F relaxes this assumption and would simplify to model E when the proportionality parameters are equal to one. In both gender models, the proportionality parameter estimates far from one (with ratios across classes 1-5 as 1: 2: 5.5: 4.6: 2 in men and 1: 0.2 :0.4 :2 :2 in women). In our case, we justified that model E fails to converge as the simultaneous modeling of the 26 parameters in model E are affected by the strong assumption that this ratio is one across all classes.

No action taken to modify the manuscript.

Minor questions:

6. Introduction: It may be good to also point out the value of the trajectory approach in better understanding the causes of between-individual variation in certain features (eg, weight variation over age), by analyzing the trajectory as an outcome rather than exposure.

Authors' reply and action: Yes, we strongly agree with this point and we have added into the Introduction and in the Discussion. Thank you.

Top of page 5:

“Thirdly, the trajectory approach allows a better understanding of the causes of between-individual variation in certain features (e.g., weight variation over age), by analysing the trajectory as an outcome rather than exposure.”

Bottom of page 14:

“This strategy is a new methodological paradigm, as the repeated measurement of a risk factor (here, BMI) becomes a clinically-relevant endpoint rather than just an exposure.”

7. Methods: the online codes are not accessible.

Authors' reply and action: The online codes have now been made publicly accessible and cited in the manuscript on page 10:

“Statistical algorithms

All R and SAS codes used to implement these tools are available via the authors and can be downloaded from www.github.com/hlennon/LCTMtools ”

8. Statistical power: should it be considered when selecting different models? Is there any effort to help researchers optimize the validity-efficiency balance when choosing the trajectory models?

Authors' reply and action: Statistical power, defined by the probability to reject the null hypothesis of a reduced nested model versus a saturated model, requires the distribution of the test statistic under the null hypothesis to be known. In a simple (one class) mixed-effects model, the likelihood ratio test (LRT) can be used for nested models and is chi-squared distributed with degrees of freedom equal to the difference in number of parameters between the two nested models.

Although, we believe that it is important to consider statistical power to select between nested models, the order of the models A-G are not nested. For example, the proportionality parameterisation of Model F is not a nested model of G, only a simplification and therefore a likelihood ratio test would not be appropriate. Therefore, when preparing the manuscript, we felt introducing the LRT to compare between different model structures (Step 3) when the class number was fixed would be cumbersome to the readers as it is may be unclear which of the Models A to G were in fact nested without introducing the equations, which is not the aim of the BMJ Open journal.

Additionally for latent class models comparing models with differing number of classes, the LRT with the likelihoods of a K-1 class and a K class model is not chi-squared distributed and therefore cannot be used to select between different models with different number of classes (Step 2).

No specific action taken.

9. The last point in the “strengths and limitations of this study” panel is very generic and not directly related to the primary content of the current study. It would be good to consider the limitations of the proposed framework rather than the substantive analysis.

Authors' reply and action: In hindsight, we agree and have revised this section extensively as follows:

STRENGTHS AND LIMITATIONS OF THIS STUDY

- We developed a systematic approach, with rationale for each of eight steps, to derive a latent class trajectory model of favoured number of classes and ‘core’ model structure specification
- The results presented here are based on modelling data from a large well-characterised US cohort; allowing the derivation of numerically meaningful subpopulations (i.e., classes) with distinct phenotypes
- Compared with ‘one-off’ BMI categorisation, latent class trajectory modelling offers additional phenotypic information and opportunities to identify and intervene early in sub-populations with adverse trajectories
- Whilst we described multiple diagnostic tests, ultimately model selection was based on case-study appropriate model interpretation (for example, model adequacy; discrimination; clinical plausibility; sensitivity analyses) by a multi-disciplinary research team

10. Page 6: the “Step 3” paragraph, the reference to Table S2 should be S1. Authors’ reply and action: Thank you. This has been corrected.

11. Ref 19 is irrelevant and can be removed.

Authors’ reply and action: Thank you. This reference has been removed.

Reviewer: #2

Reviewer Name: Maaïke Koning

Institution and Country: Windesheim University of applied Sciences

Please state any competing interests: None declared

Please leave your comments for the authors below

I find it hard to follow the main objective of this paper. The title suggests that it concerns a framework for future studies with latent class models. The main conclusion does not answer this question. Using different models is an important part of exploring data, and it would be a bit simple to choose one model and postulate this as a framework.

Authors’ reply and actions: We agree with the reviewer on the important point of exploring the principle of many models. However, in the setting of latent class trajectory modelling, there may be greater than 80 models to select from.

We have clarified and added this to the Introduction, page 5, as follows:

“For many exposures of interest, typically two to seven classes might be described and, as detailed latter, at least seven model structures might be fitted, with and without linear curve properties, such that it is possible to derive greater than eighty different models. Thus, reported differences between studies using latent class modelling might reflect different modelling assumptions rather than true differences between populations.”

We also agree with the reviewer that there is no one ideal model- and we have revised our manuscript emphasizing the need for a ‘core’ favoured model. We now clarify this in the Introduction, page 5, and elsewhere, as:

“To facilitate the generalizability of results in future studies, here, we propose a framework to construct and select a ‘core’ LCTM , using an example of repeatedly determined body mass index (BMI) across adulthood in the National Institutes of Health (NIH)-AARP Diet and Health Study cohort.”

2. The trajectories are based upon self reported BMI, and also recalled BMI from years back. These are limitations which should be mentioned.

Authors’ reply: While this is an important point, it refers mainly to biases that are relevant to the exposure -disease association component of where latent class modelling might be used – this is generally a second stage.

We have clarified this at the end of the Introduction, page 5, as follows:

“For exposure-disease outcome association analyses, current approaches generally use two stages: first, latent class trajectory modelling, followed by standard association modelling. The framework described here is limited to the first stage.”

3. Also it is mentioned that the kappa statistic may not be the best criterium to use in latent class modelling, but it is not mentioned in the methods that this value is used in any way. Furthermore I would reduce the amount of information and tables shown, for example figure FS3 seems redundant to me, it makes it difficult to find the focus of the paper. What happened to missing data?

Authors’ reply and action: There are three parts to this comment, and we address these separately.

3.1 We have clarified the use of the kappa statistic in the Methods, bottom of page 8, as follows:

“..... concordance of class membership with conventional BMI category membership using the

kappa statistic (as LCTM is an unsupervised learning approach, we computed for all possible combinations and selected the optimal).”

And then its interpretation in the Discussion, page 14, as follows:

“..... clinical characteristics derived from the LCTM differed from those derived from conventional

categorisation approaches. Thus, compared with ‘one-off’ BMI categorisation, latent class trajectory modelling offers additional phenotypic information.”

3.2 Figure S3 is a set of spaghetti plots to allow the reader the opportunity to 'get a feel for the individual-level data'. Given that reviewer #1 has raised the concern that latent class modelling has often been perceived as a 'black box', we argue that Figure S3 is both relevant to the paper and its transparency, and respectively, request that it stays.

In the original submission, in the Results page 10, we wrote:

"Spaghetti plots of individual level data illustrated that the timing and size of BMI changes characterise the classes – for example, sharp increases in BMI in early adulthood in Class III but later in adulthood for Class IV (Figure S3)."

3.3 Regarding missing data, for pragmatic reasons (as we are concentrating on methodology, here), we limited our main modelling to individuals with BMI values for all four time points.

As sensitivity analyses, we addressed missing data in our original submissions on page 11, as follows:

"We tested the favoured model using a larger sample of individuals with at least three measures, and found no material differences between these models, in men and women, and the main model (Figure S4)."

We tested for individuals with at least two measures, and models failed to converge. To keep to word count limits, we did not state this.

4. Is the paper reported according to STROBE or CONSORT guidelines? Authors' reply: Neither STROBE nor CONSORT apply here.

This paper aims to set out a framework for medical researchers to follow to investigate the model structure of latent class trajectory models.

5. It is not explained in the methods that lifestyle behaviours were investigated, yet these are presented in the tables.

Authors' reply: Thank you. We now clarify this in the methods, page 7, as follows:

"To assess the interpretability of the resulting classes, we investigated characteristics of lifestyle behaviours of the favoured model such as smoking, alcohol consumption and physical activity."

In Results, we go on to tabulate the new latent classes against the lifestyle characteristics and show these in Table S5 and S6.

6. Strengthen the choice made for building models for men and women separately. No confounding effects of gender? Also is there a difference in the outcomes between men and women?

Authors' reply and action: Thank you. We have now clarified this important point early in our Methods, page 6, as follows:

“We built models for both genders, as BMI patterns of lifetime changes differ for men and women .”
There were no disease outcomes in this study, so there was no tests for effect modification by gender.

7. Overall: the manuscript needs clarification of the main objective and what was done to come to the conclusions.

Authors' reply and action: This is an important point that we have part covered in our reply to point no. 1 from reviewer #2.

Additionally, and in line with our reply to point no. 1, we have now extensively revised our abstract with a clearer objective and conclusion as follows:

“Objectives: Latent class trajectory modelling (LCTM) is a relatively new methodology in epidemiology to describe life-course exposures, which simplifies heterogeneous populations into homogeneous patterns or classes. However, for a given dataset, it is possible to derive scores of different models based on number of classes, model structure, and trajectory property. Here, we rationalise a systematic framework to derive a ‘core’ favoured model.

.....**Conclusion:** We propose a framework to construct and select a ‘core’ LCTM, which will facilitate generalizability of results in future studies.”

Furthermore, in order to better contextualize our objective and conclusions, we have extensively revised our Future Research sections, page 14, as follows:

“For future research, improving the construction, interpretation and reporting of LCTM (advocated here) is hugely important as the LCTM approach has opportunities to identify and intervene early in sub-populations with adverse trajectories. This approach is analogous to the well-held public strategy of using childhood growth charts to identify and intervening in young children failing to thrive. Thus, in the example of BMI, remembering that 80% of obese adults were not obese in childhood, future LCTM studies might identify (new) individuals in their 20s or early 30s on adverse trajectories towards later adulthood obesity. This strategy is a new methodological paradigm, as the repeated

measurement of a risk factor (here, BMI) becomes a clinically-relevant endpoint rather than just an exposure.”

Reviewer # 3.

Reviewer Name: Nilesh Shah

Institution and Country: University of Pittsburgh, USA

Please state any competing interests: None declared

Please leave your comments for the authors below

The authors outline a systematic framework and procedure for building latent class trajectory models, and demonstrate their approach by analyzing BMI data. Given the varying results that trajectory analyses can yield given different assumptions, it is important to consider decisions that lead to optimal models. The authors lay out a process to help analysts determine how to select the optimal model for their given data. This is a good and necessary step toward standardizing these types of analyses.

Authors' reply: Thank you for this succinct summary and these positive comments.

Comments:

1. Step 3 of the procedure is not clear. It seems as though the text does not correspond with the table referenced (Table S2). The authors may be referring to Table S1.

Authors' reply and action: Thank you, we have amended the reference to Table S1

2. Please include the BIC for each of these models so that it is clear how you are choosing the optimal model.

Authors' reply: The BIC for each of the models is displayed next to each model in both Tables T2 and T3.

3. In Step 4, the authors describe a series of model adequacy tests, and mention that if these are violated, one may return to a previous step and select a new model with a higher BIC. The authors may want to discuss the trade-off between BIC efficiency and model adequacy. Is it context dependent? The authors may want to explore if the BIC Factor measure provides a better idea of k rather than BIC. How would using BIC Factor lead to different results, particularly when adjusting for many covariates? It does not appear as though the authors controlled for baseline covariates in their BMI model. How would the addition of covariates (and models with more parameters) affect the model selection process?

Authors' reply: There are three parts to this comment, and we address these separately.

3.1 As per our reply to point no. 4, reviewer #1, we have revised the manuscript in several sections to point out that BIC is not the sole criteria for model structure selection.

3.2 Thank you for your comment on whether the BIC is the most suitable for model selection. We haven't come across the BIC Factor. We have assumed Reviewer #3 is referring to the Bayes Factor, which can be approximated by the change in BIC i.e. $\approx \ln \frac{BIC_1 - BIC_2}{2}$ and can be used in this context. However, it is standard to use the BIC for this purpose. Not only does it provide an approximation to the Bayes factor, it has good empirical results for recovering the correct number of classes in latent class modelling (Lukociene and Vermunt, 2001). We have explored the Bayes factor in conjunction with the BIC, and displayed in the tables below. The conclusions drawn with the Bayes factor would be no different to using the Bayes Factor. Therefore we prefer to use the BIC.

Model	Description	BIC	Favoured model	
Men				
A	Homoscedastic	3476322		
B	Heteroscedastic	3511646	-35324	Favours A over B
C	Random intercept	3364856	146790	Favours C over B
D	Random slope	3325463	39393	Favours D over C
F	Random quadratic, Proportional	3301301	24162	Favours F over D
G	Random quadratic, Unrestricted	3320005	-18704	Favours F over G
Women				
A	Homoscedastic	2289509		
B	Heteroscedastic	2238444	51065	Favours B over A
C	Random intercept	2240681	-2237	Favours B over C
D	Random slope	2193155	47526	Favours D over C
E	Random quadratic, Equal	2188224	4931	Favours E over D
F	Random quadratic, Proportional	2169793	18431	Favours F over D
G	Random quadratic, Unrestricted	2187707	-17914	Favours F over G

Model	BIC	Favoured model
Men		

1	*		
2	3324009		
3	3310908	13101	Favours 3 class over 2
4	3324128	-13220	Favours 3 class over 4
5	3301301	22827	Favours 5 classes over 4
6	*		
Women			
1	*		
2	2195386		
3	2179080	16306	Favours 3 class over 2
4	2179137	-57	Favours 3 class over 4
5	2169791	9346	Favours 5 classes over 4
6	*		

* Failed to converge

3.3 Regarding adjustment for covariates, our aim was to derive classes based on BMI values only, therefore, so we have not adjusted for covariates.

5. In the discussion, the authors state uncertainty in class membership is not discussed, and that is a criticism of these models. Shah, et al formulate measures for evaluating discrimination and class membership in "Measures of discrimination for latent group-based trajectory models", Journal of Applied Statistics, 42(1) 2015.

Authors' reply and action: Thank you for highlighting this paper. We have added the following in the

Discussion, page 12, as follows:

"A further modification of discrimination measurement with variance estimation has been described by Shah and colleagues, and might have importance for class assignment where 'yes/no' treatment decisions are required."

6. One of the strengths outlined by the authors is the large dataset they used to demonstrate their process. Is there any idea how this process holds up with smaller sample sizes?

Authors' reply : This is a project for the future, We currently have a PhD student investigating the role of the sample size in determining the classes. We expect similar results to linear mixed models.

VERSION 2 – REVIEW

REVIEWER	Mingyang Song Harvard Medical School, USA
REVIEW RETURNED	19-Feb-2018

GENERAL COMMENTS	The authors have addressed most of my comments except for Q8 about statistical power. The fact that likelihood ratio test may not be appropriate for comparing trajectory models that are not nested does not diminish the importance of considering statistical power in model selection, since validity and efficiency are two key quality measures of any modeling practice. Because modeling is the focus of this paper, it would be useful to at least discuss the implications of balancing validity versus efficiency in building trajectory models.
---

REVIEWER	Maaïke Koning Windesheim University of Applied Sciences
REVIEW RETURNED	04-Mar-2018

GENERAL COMMENTS	I have no further comments.
-----------------------------

VERSION 2 – AUTHOR RESPONSE

Reviewer #1:

Reviewer Name: Mingyang Song

Institution and Country: Harvard Medical School, USA

Please state any competing interests: None declared

The authors have addressed most of my comments except for Q8 about statistical power. The fact that likelihood ratio test may not be appropriate for comparing trajectory models that are not nested does not diminish the importance of considering statistical power in model selection, since validity and efficiency are two key quality measures of any modeling practice. Because modeling is the focus of this paper, it would be useful to at least discuss the implications of balancing validity versus efficiency in building trajectory models.

Authors' reply and action: Thank you. Yes, we appreciate this point.

In the discussion, we now state at the bottom of page 13,

“Some discussion on statistical power and efficiency is warranted. The objective of model selection is a trade-off between efficiency and validation with the aim of summarising distinct features of the data as parsimonious as possible, and not just the maximisation of model fits (6). For example, in an hypothetical scenario, putting emphasis on the efficiency of a model in which ten classes provides the most statistically efficient model is questioned if three of the ten classes each includes less than 0.5% of the population, and without markedly different characteristics.”

VERSION 3 – REVIEW

REVIEWER	Mingyang Song Harvard Medical School, USA
REVIEW RETURNED	17-Mar-2018

GENERAL COMMENTS	The authors have added a few lines to the discussion about validity vs. efficiency. But there are a couple of misstatements. First, it should be "validity" rather than "validation" in the sentence "a trade-off between efficiency...". Second, in the hypothetic scenario the authors provided, the emphasis should actually be "validity" rather than "efficiency", because a model with more classes tend to provide higher validity (ie, better model fit) but lower efficiency due to limited power in deriving the classes. Thus, please revise the description. Here is what I would suggest: "For example, in a hypothetical scenario, putting TOO MUCH emphasis on the VALIDITY of a model in which ten classes provide the BEST model FIT is questionABLE if three of the ten classes each include less than 0.5% of the population and DO NOT SHOW markedly different characteristics".
--

VERSION 3 – AUTHOR RESPONSE

Reviewer #1:

Reviewer Name: Mingyang Song

Institution and Country: Harvard Medical School, USA

Please state any competing interests: None declared

The authors have added a few lines to the discussion about validity vs. efficiency. But there are a couple of misstatements. First, it should be "validity" rather than "validation" in the sentence "a tradeoff

between efficiency...". Second, in the hypothetic scenario the authors provided, the emphasis should actually be "validity" rather than "efficiency", because a model with more classes tend to provide higher validity (ie, better model fit) but lower efficiency due to limited power in deriving the classes. Thus, please revise the description. Here is what I would suggest: "For example, in a hypothetical scenario, putting TOO MUCH emphasis on the VALIDITY of a model in which ten classes

provide the BEST model FIT is questionABLE if three of the ten classes each include less than 0.5% of the population and DO NOT SHOW markedly different characteristics".

Authors' reply and action: Thank you. In the discussion, we now state at the bottom of page 13, "For example, in a hypothetical scenario, putting too much emphasis on the validity of a model in which ten classes provide the best model fit is questionable if three of the ten classes each include less than 0.5% of the population and do not show markedly different characteristics.